# Interactive Rationale Extraction for Text Classification

**Jiayi Dai**
Department of Computing Science
University of Alberta
dai1@ualberta.ca

**Mi-Young Kim**
Department of Science
University of Alberta
miyoung2@ualberta.ca

**Randy Goebel**
Department of Computing Science
University of Alberta
rgoebel@ualberta.ca

## Abstract

Deep neural networks show superior performance in text classification tasks, but their poor interpretability and explainability can create trust issues. For text classification problems, the identification of textual sub-phrases or "rationales" is one strategy for attempting to find the most influential portions of text, which can be conveyed as critical in making classification decisions. Selective models for rationale extraction faithfully explain a neural classifier's predictions by training a rationale generator and a text classifier jointly: the generator identifies rationales and the classifier predicts a category solely based on those rationales. The selected rationales are then viewed as the explanations for the classifier's predictions. Through exchange of such explanations, humans can interact to achieve higher performances in problem solving. To imitate the interactive process of humans, we propose a simple interactive rationale extraction architecture that selects pairs of rationales and then makes predictions from two independently trained selective models. We show how this architecture outperforms both base models for text classification tasks on datasets *IMDB movie reviews* and *20 Newsgroups* in terms of predictive performance.

[1]

## 1   Introduction

Selective (or select-predict) models for rationale extraction in text classification (Lei et al., 2016; Bastings et al., 2019), with the general structure shown in Figure 1, are designed to extract a set of words, namely a *rationale* (Zaidan et al., 2007), from an original text. For prediction purposes, the rationale is expected to be *sufficient* as the input for the classification model to obtain the same prediction as that prediction was based on the whole text. For the purpose of interpretability, the rationale should be *concise* and *contiguous*. A rationale extraction model is *faithful* (Lipton, 2018) if the extracted rationales are truly the information used for classification (Jain et al., 2020). The problem of extracting rationales that satisfy the criteria above is challenging from a machine learning perspective and becomes more difficult with only instance-level supervision (i.e., without token-level annotations) (Jain et al., 2020). One model's identification of rationales can suffer from high variance because of the complex training process. An ensemble of more than one model helps to reduce variance, which leads to the exploration of *how to make use of two rationale extraction models and how to make a choice when the two models make different predictions*.

---

[1]The implementation is provided on https://github.com/JiayiDai/RationaleExtraction.

2022 Trustworthy and Socially Responsible Machine Learning (TSRML 2022) co-located with NeurIPS 2022.

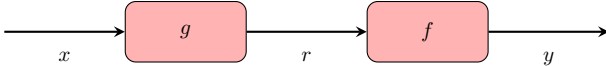

Figure 1: Schematic of selective rationale extraction models where $x$ is an embedded text, $g$ is a generator and $f$ is a classifier. Generator $g$ extracts a rationale $r$ based on which classifier $f$ makes a prediction $y$.

When two humans have different answers to a prediction, they tend to exchange their reasons or explanations, after which there might be a change of mind. To show why this interaction of humans is effective, we use the problem of proving a mathematical conjecture as an instance: because searching for a correct mathematical proof, which then leads to a correct claim about the conjecture, is usually much more difficult than verifying a proof (e.g., $\mathcal{P} \subseteq \mathcal{NP}$ in computation theory); But often one who is not capable of finding a good proof can tell if a given proof is good. So in considering the complexity for a generator to search among all possible rationales with only remote instance-level supervision, the work of rationale extraction can be much more difficult than classification.

We may then consider alternative selective models for rationale extraction to be naturally compatible with the interactive pattern of humans. We can do this by viewing the rationales extracted by a generator as the proofs for the decisions of its classifier, which means the interaction between two base models can be performed by the exchange of their rationales. Subsequently, the problem becomes how to decide if a rationale is good or not so that we know which pairs of rationale and prediction are appropriate choices when two base models make different predictions. A *good rationale* here is expected to give a correct prediction when input to a decent classifier.

Intuitively, a good rationale is supposed to contain strong indicators for the correct "gold label" instead of insignificant words which do not contribute to classification. This leads to two simple rules for handling base models' disagreements: first, a good rationale is more likely to produce consistent predictions among classifiers (i.e., a good explanation convinces people); second, a good rationale is more likely to produce a higher *confidence level* (Section 2.2) for the prediction of one classifier (i.e., one with a good reason is often confident). The two rules create a basis for building classification models, as opposed to random guessing based on otherwise randomly selected words. Note that the two rules are based on the assumption that the probability that base models extract strong indicators for wrong labels is very low, which should be considered to be true for decent generators and decent classifiers (i.e., better than random guessing).

To imitate the interactive pattern of humans in problem solving, we introduce **Interactive Rationale Extraction for Text Classification** to interactively connect two independently trained selective rationale extraction models. We show that the architecture achieves higher predictive performance than either base models with similar performance on *IMDB movie reviews* and *20 Newsgroups*. This is done by selecting pairs of rationale and prediction from the base models using the above simple rules. Because the rationales in our architecture are generated before any classification is made, the interactive process also differs from post-hoc processing, such as Lime (Ribeiro et al., 2016), which generates a rationale or token-level importance scores after making a prediction. In addition, because this interactive architecture makes decisions solely based on the base models' rationales, the faithfulness and interpretability of the base models' rationales are not compromised.

## 2 Background

### 2.1 Selective Rationale Extraction

The original selective rationale extraction model was proposed by Lei et al. (2016) with an architecture shown in Figure 1. Their model faithfully explains a neural network-based classifier's predictions by jointly training a generator and a classifier with only instance-level supervision. We summarize their work as follows. The generator $g$ consumes the embedded tokens of the original text, namely $x = [x_1, x_2, ..., x_l]$ where $l$ is the number of the tokens in the text and each token $x_i \in \mathbb{R}^d$ is an $d$ dimensional embedding vector, and outputs a probability distribution $p(z|x)$ over the hard mask $z = [z_1, z_2, ..., z_l]$ where each value $z_i \in \{0, 1\}$ denotes whether the corresponding token is selected. A rationale $r$ is defined as $(z, x)$ representing the hard mask $z$ over the original input $x$. Subsequently, the classifier $f$ takes $(z, x)$ as input to make a prediction $f(z, x)$. Given gold label $y$, the loss function

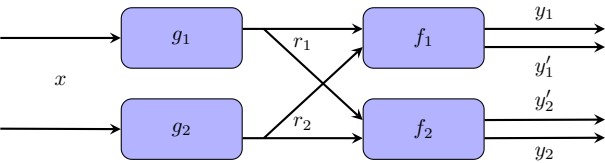

Figure 2: Schematic of our interactive rationale extraction where rationales are exchanged. The notations follow Figure 1.

used to optimize both generator $g$ and classifier $f$ is defined as

$$loss(z, x, y) = ||f(z, x) - y||_2^2 + \lambda_1 ||z|| + \lambda_2 \sum_{i=1}^{l-1} |z_i - z_{i+1}| \qquad (1)$$

which consists of three parts: prediction loss, selection loss and contiguity loss. The parameters $\lambda_1$ and $\lambda_2$ in the loss function are used to tune the constraints on rationales (i.e., conciseness and contiguity). Jain et al. (2020) modified the loss function to apply hard constraints on rationales (i.e., maximum length) by not punishing a model when a given limit on the number of words is not reached.

Because of the absence of token-level supervision and the use of hard masking which is not differentiable, Lei et al. (2016) turned to REINFORCE (Williams, 1992) for gradient estimation, which causes high variance and sensitivity to hyper-parameters (Jain et al., 2020). Following the select-predict architecture proposed by Lei et al. (2016), Bastings et al. (2019) explored a re-parameterization heuristic called HardKuma for gradient estimation. Furthermore, Guerreiro and Martins (2021) exposed the trade-off between differentiable masking and hard constraints in selective rationale extraction models.

## 2.2 Confidence Level

Confidence level (CL) is a measure that indicates how far a neural network's prediction is from being neutral. Given a neural network's non-probabilistic output $k = [k_1, k_2, ..., k_n]$ for a $n$-class classification, Kumar et al. (2022) defined the CL of the classification with a softmax function

$$CL(k) = \frac{exp(max(k))}{\sum_{i=1}^{n} exp(k_i)} \qquad (2)$$

where $max(k)$ is the value of the output node $k_i$ with the highest value (i.e., $i$ is the final prediction).

Guo et al. (2017) stated that a classification network should not only have a high accuracy but also indicate how likely each prediction is correct or incorrect for trust purposes. In addition, their study on neural networks' calibration Guo et al. (2017) suggested that accuracy, even if not nearly identical to CL for some neural networks, is generally positively correlated to CL. This means that, when two base models with similar expected performances make different predictions, the prediction with a higher CL is generally more likely to be correct.

## 3 Algorithm

As demonstrated in Figure 2, after the interaction between two base select-predict models, a total of 4 predictions are generated: $y_1 = f_1(r_1)$, $y_1' = f_1(r_2)$, $y_2' = f_2(r_1)$ and $y_2 = f_2(r_2)$ where $y_1$ and $y_2$ are the predictions based on their own rationales and $y_1'$ and $y_2'$ are predictions based on the exchanged rationales, as shown in the table below.

|       | $r_1$  | $r_2$  |
|-------|--------|--------|
| $f_1$ | $y_1$  | $y_1'$ |
| $f_2$ | $y_2'$ | $y_2$  |

Given an input text, when the predictions of two base models are the same, namely $y_1 = y_2$, both rationales $r_1, r_2$ are good and the final prediction is the shared prediction. When two base models

initially show a disagreement, we check if one rationale causes more consistent predictions. If $r_1$ causes more consistent predictions, in order words, if $r_1$ changes the prediction of $f_2$ to $y_1$ when given as an input rationale (namely, $y_1 = y_2'$), but $r_2$ does not change the prediction of $f_1$ to $y_2$ when given as an input rationale ($y_2 \neq y_1'$), then the pair $(r_1, y_1)$ is chosen as the final rationale and prediction; symmetrically, if $r_2$ causes more consistent predictions, the pair $(r_2, y_2)$ is chosen. For the cases where no rationale causes more consistent predictions, we rely on confidence levels which are real numbers between 0 and 1 as defined by expression (2). If the confidence level of $f_1$ on $r_1$ is higher than that of $f_2$ on $r_2$ (say $CL(f_1, r_1) > CL(f_2, r_2)$ with $(f_1, r_1)$ and $(f_2, r_2)$ separately denoting their corresponding non-probabilistic outputs), the pair $(r_1, y_1)$ is chosen; otherwise, the pair $(r_2, y_2)$ is chosen. The process of selecting a pair of rationale and prediction is formalized in Algorithm 1. It's worth mentioning that, in implementation, the exchange of rationales only needs to be performed when base models have a disagreement in prediction (i.e., $y_1 \neq y_2$).

---

**Algorithm 1** Rationale-prediction Selection after Interaction

---

**Require:** $f_1, f_2, r_1, r_2, y_1, y_1', y_2', y_2$ from Figure 2, $CL(f, r)$ for the confidence level of $f$ on $r$.
  **if** $y_1 = y_2$ **then**                                                                 ▷ agreement
      return $(r_1, y_1)$                                                       ▷ or $(r_2, y_2)$
  **else**                                                                 ▷ disagreement
      **if** $y_1 = y_2'$ and $y_2 \neq y_1'$ **then**                ▷ model 2 convinced by model 1
          return $(r_1, y_1)$
      **else if** $y_1 \neq y_2'$ and $y_2 = y_1'$ **then**         ▷ model 1 convinced by model 2
          return $(r_2, y_2)$
      **else**
          **if** $CL(f_1, r_1) > CL(f_2, r_2)$ **then**         ▷ model 1 is more confident
              return $(r_1, y_1)$
          **else**                                           ▷ model 2 is more confident
              return $(r_2, y_2)$
          **end if**
      **end if**
  **end if**

---

# 4 Experiments

## 4.1 Datasets

**IMDB movie reviews (Maas et al., 2011)**    This is a dataset of 50,000 movie reviews collected from the Internet Movie Database (IMDB) with binary labels (i.e., positive and negative). The dataset is originally split into two subsets: 25,000 for training and 25,000 for testing. We randomly split the training data into 20,000 (80%) for training and 5,000 (20%) for development. The two labels are perfectly balanced in each subset.

**20 Newsgroups**    It is a publicly available dataset containing a total of 18,846 texts, with 11,314 for training and 7,532 for testing, in 20 distinct categories of news topics. We split the training data randomly into 9,051 (80%) for training and 2,263 (20%) for development. The numbers of the 20 labels are not perfectly balanced and varying from 304 to 490 in the training data, 73 to 131 in the development data and 251 to 399 in the testing data.

## 4.2 Setup

**Training**    Instead of REINFORCE (Williams, 1992), a re-parameterization heuristic Gumbel-Softmax (Jang et al., 2017) is used to simplify gradient estimation. A convolutional neural network (Kim, 2014) is used for both generators and classifiers with filter sizes of [3,4,5], filter number of 100 and dropout rate of 0.5. Hidden dimensions of 100 and 120 are separately used for the first and the second base model, which is the only difference among all parameters for training two base models. Adam is used as the optimizer with a weight decay of 5e-06 and an initial learning rate of 0.001. If no improvement is achieved in loss in development dataset from the previous best model after a patience of 5 epochs, the learning rate is halved (i.e., 0.001, 0.0005...) and the training process starts

| 20 Newsgroups | | | | |
|---|---|---|---|---|
| $(\lambda_1, \lambda_2)$ | (5e-3, 0) | | (1e-3, 1e-3) | |
| Base Model | Model 1 | Model 2 | Model 1 | Model 2 |
| Length | 11.33 | 11.18 | 21.76 | 22.68 |
| Contiguity Loss | 17.12 | 16.84 | 21.92 | 21.45 |
| Interaction Cases | (331, 363, 1129, 1211.5) | | (228.6, 264, 974.2, 1075.8) | |
| Case Accuracy | (0.41, 0.43, 0.30, 0.26) | | (0.38, 0.44, 0.31, 0.27) | |
| IMDB movie reviews | | | | |
| $(\lambda_1, \lambda_2)$ | (1e-3, 0) | | (2e-4, 2e-4) | |
| Base Model | Model 1 | Model 2 | Model 1 | Model 2 |
| Length | 13.99 | 17.59 | 29.22 | 27.37 |
| Contiguity Loss | 21.84 | 26.45 | 37.14 | 35.48 |
| Interaction Cases | (855.6, 946.0, 1187.4, 1250.0) | | (681.7, 665.2, 1101.8, 1295.7) | |
| Case Accuracy | (0.66, 0.65, 0.59, 0.59) | | (0.66, 0.64, 0.58, 0.60) | |

Table 1: Experiment details (average values). We report the rationale length (i.e., number of words) and contiguity loss of each base model under each hyper-parameter setting and numbers of interaction cases and each case's accuracy. Four values in an interaction case are the average numbers of the cases separately for base model 1 convinced, base model 2 convinced, base model 1 more confident, and base model 2 more confident. These are the four cases of handling disagreements in Algorithm 1.

over from the previous best model. In total, 20 epochs are used for training. Cross-entropy is used as the loss objective. Batch size is set to be 128. For Gumbel-Softmax (Jang et al., 2017), the initial temperature is 1 with a decay rate of 1e-5. GloVe (Pennington et al., 2014) of embedding dimension 300 is used for word embedding. The max text lengths are separately set to be 80 and 200 words for *20 Newsgroups* and *IMDB movie reviews*.

**Testing**   For each dataset, two base models are trained and tested with two settings of hyper-parameters $(\lambda_1, \lambda_2)$ from the loss function, $\{(0.005, 0), (0.001, 0.001)\}$ for *20 Newsgroups* and $\{(0.001, 0), (0.0002, 0.0002)\}$ for *IMDB movie reviews*. The four settings are chosen in a way that shows the performance of the algorithm under different rationale length and contiguity (Table 1). For each hyper-parameter setting, both base models are trained and tested with 6 random seeds (i.e., {2022, 2023, 2024, 2025, 2026, 2027}), and the invalid cases where two base models show a significant difference in the performance in development dataset (i.e., > 3% in accuracy) are removed. The numbers of invalid cases are separately 2, 1, 1, 0 out of 6 for the four hyper-parameter settings.

| | 20 Newsgroups | | IMDB movie reviews | |
|---|---|---|---|---|
| $(\lambda_1, \lambda_2)$ | (5e-3, 0) | (1e-3, 1e-3) | (1e-3, 0) | (2e-4, 2e-4) |
| Model 1 | .55 (.53-.57) | .58 (.56-.59) | .81 (.80-.82) | .82 (.81-.83) |
| Model 2 | .54 (.52-.57) | .57 (.55-.59) | .81 (.80-.82) | .82 (.81-.83) |
| Interaction | **.58 (.56-.60)** | **.60 (.59-.61)** | **.83 (.82-.84)** | **.84 (.83-.84)** |

Table 2: Average performances (accuracy) of maximum six experiments for base (Models 1 and 2) and interactive models under each hyper-parameter setting for each dataset. The (min, max) performances of each model are also reported to demonstrate variances.

## 4.3   Quantitative Evaluation

For quantitative evaluation, we report the predictive performances of the classifiers from the base models and the interactive model. In Table 2, the interactive model outperforms the better base model by 2% in *IMDB movie reviews* and 2-3% in *20 Newsgroups* and shows a relatively smaller variance in both datasets. The improvement in predictive performance and reduced variance is general for most experiments in addition to the four settings. We found that, in the cases of extreme hyper-parameter settings where rationales contain almost whole texts or no words, there is no improvement. This seems reasonable as, when base models generate rationales of whole texts or no words, the rationales are identical, which makes the exchange of rationales meaningless. Also, in some cases where one base model is trained well and one is not (e.g., 80% and 60% accuracy in *IMDB movie reviews*), the interactive model shows a slightly lower performance than the better base model. The reason can be

that a relatively better rationale generated by the better model can not convince the classifier of the poor performance model, where the first rule that a good rationale is more likely to produce consistent predictions is not followed. If no rationale is causing consistent predictions, the second rule about confidence level is applied but a poor classifier can sometimes be overconfident, which causes errors.

For a binary classification task, when two base models with similar performances have a disagreement, the expected accuracy of each base model is around 50% and the probability of blindly choosing a prediction turning out to be correct should also be near 50% (i.e., random guessing). However, as shown in Table 1, in *IMDB movie reviews*, the accuracy after interaction is 8-16% higher than random guessing. This result indicates that the interactive method performs better than majority voting because voting can not effectively handle the cases where two base models have disagreements (i.e., random guessing would be performed if two base models have a disagreement in majority voting).

Also, we observed that, when the constraints on rationales are less strict (i.e., allowing more words and more contiguity loss), generally the performance of base models increases but the improvement after interaction deceases. The reason may be that, with weaker rationale constraints, strong indicators are easier identified as causing the rationales of both base models to contain more similarly strong indicators.

## 5 Conclusion

To handle the high variance of selective rationale extraction models, we proposed method we call **Interactive Rationale Extraction for Text Classification**, which selects rationales and predictions from base models based on simple rules through imitating the interaction process between humans for handling disagreements. The experimental results show that the interactive process is effective in terms of improving performance, choosing a better rationale, and reducing variance.

## Acknowledgments and Disclosure of Funding

Adam Yala provided the implementation of base selective rationale extraction models with Gumbel-Softmax in the GitHub repository `https://github.com/yala/text_nn`, which greatly saves the implementation time for our experiments. This research was supported by the Alberta Machine Intelligence Institute (AMII) and the Canadian Natural Sciences and Engineering Research Council (NSERC) [including funding reference numbers RGPIN-2022-03469 and DGECR-2022-00369].

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
