# OpenReview forum: "Interactive Rationale Extraction for Text Classification"
_NeurIPS.cc/2022/Workshop/TSRML — TSRML2022_

### Official Review · Reviewer_dkkb · 2022-10-20
**An intuitive way to leverage rationales by exchanging that brings performance gain**

**Overall Rating:** 8

**Summary:**

The paper aims to improve the performance of the extract-and-predict pipeline for interpretable text classification. To achieve that, the paper proposes to exchange the rationales from two base models and make decisions based on 1) whether one of the exchanged rationales changes the prediction of one classifier or 2) whether one rationale leads to a more confident prediction. Experiments show that the proposed method improves the performance of the base models on two text-classification datasets. Further analysis shows that exchanging rationales is more effective when rationales are constrained to be more concise.



**Strengths:**

1. The paper proposes an interesting method to better leverage the rationales for prediction. The method aligns well with how human might benefit from exchanging rationales for prediction.
2. Experiments show non-trivial improvement over the baselines, validating the effectiveness of the proposed method.

**Weaknesses:**

1. The proposed method can be viewed as a way to ensemble different models. Thus, some vanilla baselines such as majority voting should be considered as baselines.

**Overall Recommendation:**

Overall I recommend accepting the paper given the non-trivial idea and improvement presented in the paper.

**Review Confidence:**

5: The reviewer is absolutely certain that the evaluation is correct and very familiar with the relevant literature

---

### Official Review · Reviewer_vWc7 · 2022-10-21

**Overall Rating:** 4

**Summary:**

This paper proposed an interactive rationale extraction architecture for text classification problems. The motivation of this approach is to imitate the communication process between humans in problem solving. Authors utilized two independently trained base select-predict models as the base models. When base models have a disagreement in prediction, an interactive process is incorporated to exchange their selected rationales and further produce exchanged predictions. By considering consistency in prediction and models’ confidence, a final selection can be made.


**Strengths:**

The motivation of this paper is clearly clarified and the methodology is easy to understand.


**Weaknesses:**

- The novelty is limited because the interactive process of two base models can be seen as post-hoc processing with rule based decision making, which is similar to bagging. Hence It would be more interesting if interaction can be incorporated into ensemble training.

- It seems two base models are almost the same besides hidden dimension, so it would be better to investigate the influence of using two different base models. Also, I am interested in how the number of base models can influence the decision making.


**Overall Recommendation:**

Although the motivation of this paper is clearly clarified, the methodology needs to be improved from post-hoc processing.

**Review Confidence:**

3: The reviewer is fairly confident that the evaluation is correct

---

### Decision · Program_Chairs · 2022-10-23

**Decision:**

Accept

**Comment:**

The manuscript is accepted based on the simple but empirically effective method it proposed. Both reviewers mention that the manuscript lacks discussion about its relationship with ensemble training. Can it be seen as post-hoc processing? How does it compare with simple voting or bagging ensemble baselines? We strongly encourage the authors to take these comments into consideration in the camera-ready version.